# CHARACTER MIXING FOR VIDEO GENERATION

**Prompts-1:** *Ice Bear* calmly paints a picture of **Tom**, while **Tom** keeps trying to pose but falls into the paint buckets.
**Prompts-2:** *Mr. Bean* blows up a balloon. **Jerry** hides inside. When the balloon pops **Jerry** lands on **Mr. Bean's** head
**Prompts-3:** *Young Sheldon* judges a spelling bee, **Panda** spells words wrong on purpose, while **Jerry** sneaks in funny answers.

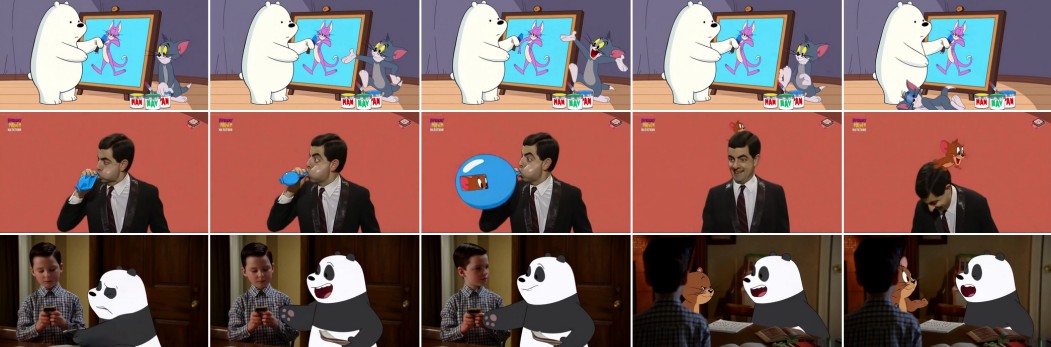

Figure 1: **Multi-character Mixing.** Our method preserves character identity, behavior and original style while generating plausible interactions between characters that have never coexisted—from cartoons (*We Bare Bears*, *Tom and Jerry*) to realistic humans (*Mr. Bean*, *Young Sheldon*).

## ABSTRACT

Imagine Mr. Bean stepping into *Tom and Jerry*—can we generate videos where characters interact naturally across different worlds? We study inter-character interaction in text-to-video generation, where the key challenge is to preserve each character's identity and behaviors while enabling coherent cross-context interaction. This is difficult because characters may never have coexisted and because mixing styles often causes ***style delusion***, where realistic characters appear cartoonish or vice versa. We introduce a framework that tackles these issues with Cross-Character Embedding (CCE), which learns identity and behavioral logic across multimodal sources, and Cross-Character Augmentation (CCA), which enriches training with synthetic co-existence and mixed-style data. Together, these techniques allow natural interactions between previously uncoexistent characters without losing stylistic fidelity. Experiments on a curated benchmark of cartoons and live-action series with 10 characters show clear improvements in identity preservation, interaction quality, and robustness to style delusion, enabling new forms of generative storytelling. Our project page *https://mi-mi-x.github.io*.

## 1 INTRODUCTION

In an era where films and iconic characters are just a click away, a natural question arises: what if we could unite these beloved characters together, merging their roles and interactions into a single story?

Since the release of Sora (OpenAI, 2024) by OpenAI, fundamental text-to-video (T2V) generation models (Yang et al., 2024; OpenAI, 2024; Team Wan, 2025; Team, 2024b; DeepMind, 2024; Kong et al., 2024; kli, 2024; Team Wan, 2025) have achieved substantial progress in general video synthesis. For producing videos centered on specific or customized characters, a common approach is to leverage a reference image as input. Such personalized video generation methods (He et al., 2024; Liang et al., 2025; Jiang et al., 2024; Wei et al., 2024; Fei et al., 2025; Chen et al., 2025c) allow users to create customized content with their own images.

However, references images alone are not sufficient to reveal the character's unique behaviors and how they interact with the environment and with one another. Thus, while preserving the identity, these approaches often fail to faithfully capture the character's complex behavior. To this end, we want to develop a method that could allow the model to learn from all the footages and scripts of the characters of interest that can be collected, so as to learning not only their appearance but also motion idiosyncrasies and personality, and to generate vivid videos that maximally match their behavior traits and motion patterns.

To faithfully generate videos for a single character, one can fine-tune a text-to-video model on footage of that specific character. However, moving beyond single-character settings brings two major challenges.

The first is the **non-coexistence challenge**: characters from different shows never co-occur in any training video, leaving no paired data to model their joint interactions. To address this, we explicitly encode each character's identity and behavior into text by annotating their names and actions in the captions. This disentangles character-specific behavior embeddings from the underlying training videos, enabling us to fine-tune one foundation model on each character's individual footages while still allowing them to co-exist and interact naturally at inference time.

The second is the **style delusion challenge**: characters often originate from domains with drastically different visual styles, such as live-action sitcoms and cartoons, which never naturally co-exist in the same video. Directly training on mixed styled data leads to unstable character styles, as shown in Figure 2.

We tackle this by introducing a style-aware data augmentation strategy that composites characters from different domains into the same video while preserving their native appearances. We find that even a small proportion of such augmented data substantially improves style preservation in cross-domain character mixing. For background ambiguities, we introduce an extra prompt for background style.

Figure 2: **Style delusion examples.** When mixing different style characters, their styles may shift undesirably. For instance, Mr. Bean looks cartoonish (top row), while Ice Bear appears realistic (bottom row).

To validate our approach, we curate an 81-hour (52,000 clips) dataset featuring two cartoons (*Tom and Jerry*, *We Bare Bears*) and two realistic shows (*Young Sheldon*, *Mr. Bean*). Each clip is annotated with explicit character names and style information, supporting fine-grained control during training and inference. We further establish the first benchmark for multi-character video generation, evaluating identity preservation, motion fidelity, interaction realism, and style consistency.

Our contributions are summarized as follows:

- We proposed the first video generation framework for multi-character mixing that addresses both the non-coexistence challenge and the style dilution challenge.

- We curated an 81-hour (52,000 clips) dataset with character- and style-annotated captions, enabling controllable multi-character video synthesis across domains.

- We conducted extensive experiments and introduced a benchmark, showing that our method significantly improves identity preservation, motion consistency, and interaction quality compared to prior art.

## 2 RELATED WORKS

**Video Generation** The advent of diffusion models has transformed video generation, advancing from early text-to-video systems such as ImagenVideo (Ho et al., 2022), Make-A-Video (Singer et al., 2022), and VideoLDM (Blattmann et al., 2023), to large-scale architectures like Sora (OpenAI, 2024b), Goku (Chen et al., 2025b), Wan2.1 (Team Wan, 2025) and HunyuanVideo (Kong et al., 2024), which achieve state-of-the-art general video synthesis. However, these models remain limited in generating content with specific identities or custom subjects, motivating the emerging direction

of personalized video generation, where reference visual signals guide the synthesis of videos with consistent appearance and motion dynamics.

**Single-Concept Personalization** Early personalized video generation methods, such as DreamVideo (Wei et al., 2024), Magic-Me (Ma et al., 2024), and PersonalVideo (Li et al., 2024), customized videos with per-subject tuning, achieving identity preservation but requiring costly optimization. More recent zero-shot approaches like ID-Animator (He et al., 2024) leverage facial adapters to enable identity-consistent generation from a single reference image without fine-tuning. Despite these advances, existing methods largely emphasize visual similarity while overlooking motion-related aspects such as unique behaviors and environment interactions.

**Multi-Concept Customization** Compared to sigle-concept personalization, multi-concept customization is more challenging due to identity blending, where multiple characters risk being fused into a composite scene. Video Alchemist (Chen et al., 2025c) addresses this through cross-attention–based fusion of text and image representations for open-set subject and background control, while Movie Weaver (Liang et al., 2025) employs tuning-free anchored prompts to preserve distinct identities. Other approaches, such as CustomVideo (Wang et al., 2024) and Custom Diffusion (Kumari et al., 2023), explore parameter-efficient fine-tuning and joint optimization for multi-subject composition. However, existing image-guided methods cannot leverage video and textual data during training, limiting their ability to model realistic interactions and dynamics across characters.

## 3 METHODS

The goal of our method is to learn the essence of characters from large collections of video data and enable them to interact seamlessly in new, mixed contexts. Given the abundance of video series—spanning cartoons and live-action shows—our approach seeks to (1) capture each character's unique identity and behavioral traits, and (2) enable flexible mixing of characters across styles and universes.

Our curated dataset (Section 3.3) consists of TV shows and animations. We leverage not only video clips but also audio and scripts, which provide crucial cues about each character's personality and behavioral logic. Each domain features one or multiple central characters, sometimes appearing in ensembles (e.g., Tom and Jerry), other times in isolation (e.g., Mr. Bean).

In this section, we develop a novel training scheme that introduces **Cross-Character Embedding (CCE)** and **Cross-Character Augmentation (CCA)** to achieve robust identity modeling, behavior preservation, and style-controllable mixing.

### 3.1 CROSS-CHARACTER EMBEDDING (CCE)

Faithfully reproducing an authentic character requires learning from dynamic data that reflects not only appearance but also behavior, idiosyncratic motion patterns, and contextual habits. Static images are insufficient, as they omit the motion and interaction cues that define a character's identity.

We therefore design a framework to learn the character concept embeddings across different domains. We need to tackle Character disentanglement in multi-character shows (e.g., *Tom and Jerry*, *We Bare Bears*), where multiple identities must be separated within the same clip. We also face the challenge of **Non-coexistence** of characters from different universes, who never appear together in the training data but must interact coherently at inference.

**Character–Action Prompting.** Our key insight is to design a character–action captioning format that explicitly grounds each character's identity while separating it from scene context. Unlike standard captions that describe only visual events, our captions follow the format:

*[Character: <name>], <action>. [Character: <name>], <action>.*

This design ensures that embeddings encode characters as independent entities with disentangled actions and identities. During inference, the same prompting scheme enables coherent composition of characters who never coexisted in training—for example, Mr. Bean interacting with Tom and Jerry. Although Mr. Bean and Jerry never meet in the dataset, the model has observed how each interacts

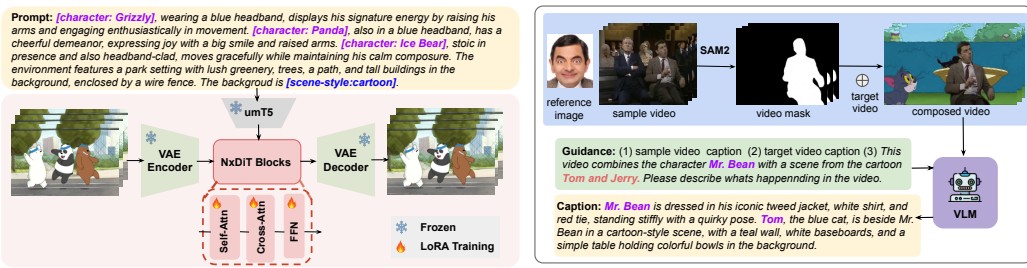

Figure 3: **Finetuning and data augmentation pipeline.**

with others in their respective domains, which generalizes to cross-universe interactions. A more detailed example can be found in Figure 3

**Prompt Generation**   We employ GPT-4o (OpenAI, 2024a) to automatically generate captions. For each short clip, we provide: 1) 10 sampled frames as visual context, 2) dialogue transcripts from the audio, and 3) source metadata (cartoon vs. TV series).

Scripts and plot summaries are also supplied to resolve ambiguity when a short clip lacks context. This setup enables GPT-4o to reliably identify character names and actions while minimizing hallucinations. The resulting process yields  52,000 video–caption pairs. Each character mention is annotated with [character:name] tags, which act as identity anchors and support controllable character-level generation. More details are in the supplemental.

**Model Adaptation**   We fine-tune Wan2.1-T2V-14B (Team Wan, 2025) with Low-Rank Adaptation (LoRA) (Hu et al., 2022). Our approach is model-agnostic and applicable to any text-to-video backbone. The curated text–video pairs capture each character's actions, emotions, and contextual behaviors, enabling the learned embeddings to serve as flexible building blocks for multi-character generation.

### 3.2   CROSS-CHARACTER AUGMENTATION (CCA)

While CCE ensures characters act authentically, training on mixed domains (cartoon vs. live-action) introduces a **style delusion problem**: characters may drift into unintended styles (e.g., *Mr. Bean* rendered as a cartoon, or Ice Bear appearing too realistic), and background styles may become unpredictable.

To preserve original styles while allowing cross-style interactions, we introduce **Cross-Character Augmentation (CCA)**. This tackles a second non-coexistence challenge: in the training data, cartoon and real characters never appear together, nor do their backgrounds.

**Synthetic Cross-Domain Compositing.**   Our intuition is that even imperfect synthetic co-occurrences can guide the model toward style-preserving generation. We therefore create augmented training clips by segmenting characters from source videos and pasting them into backgrounds from the opposite style domain. For example, Mr Bean (live-action) may be placed into a cartoon *Tom and Jerry* scene as shown in the right part of Figure 3.

Characters are segmented using SAM2 (Ravi et al., 2024), which handles both live-action and animation. To ensure relevance: For *Mr. Bean* and *Young Sheldon*, we filter clips via reference-image matching. For cartoons, we use Gemini (Team, 2024a) for automated detection and filtering.

The composited clips are then captioned by GPT, which is provided with both the background source and the inserted character identities. Each caption is further enriched with explicit style tags ([scene-style:cartoon] or [scene-style:realistic]), giving the model clear supervision for style control. The complete caption becomes

*[Character: <name>], <action>. [Character: <name>], <action>. <scene-style>*

**Prompt:** *Tom, and **Panda** go fishing on a rowboat. **Tom** keeps falling into the water while chasing his bait, **Panda** takes selfies with each fish, and **Tom** somehow catches a boot, a sandwich, and a lawn chair—but no fish.*

**Prompt:** ***Tom** and **Ice Bear** get jobs at a bakery. **Tom** keeps chasing **Jerry** through cake trays, and **Ice Bear** calmly decorates a three-tier wedding cake. The bride ends up choosing **Ice Bear**'s version over the original.*

**Prompt:** ***Mary Cooper** and **Panda** host a tea party. **Mary** brings her finest china and proper etiquette. **Panda** makes cute cupcakes with bear faces.*

**Prompt:** ***Tom** and **Sheldon** visit an aquarium. **Tom** gets mesmerized by the fish and tries to pounce on the glass. **Sheldon** gives lectures to confused toddlers about bioluminescent jellyfish.*

Figure 4: **Comparison on multi-subject interaction.** Results from SkyReel-A2 (Fei et al., 2025) (top row) and ours (bottom row).

**Empirical Findings.** We observe that a **small proportion** of such augmented clips suffices to unlock robust cross-style composition. Excessive synthetic data, however, degrades realism and harms overall video quality. A detailed analysis is provided in the experiments section.

## 3.3 TRAINING AND DATA

During fine-tuning, backbone parameters are frozen and only LoRA layers are updated, ensuring efficiency and reducing overfitting. We adopt rank-32 LoRA layers and train for 5 epochs with the Adam optimizer (learning rate 1e-4, batch size 64). Gradient clipping is applied for stability, and mixed-precision (FP16) training is used for efficiency. All experiments are conducted on NVIDIA A100 GPUs.

**Scenes and segments.** We curate a dataset comprising two cartoons and two live-action shows: approximately 9 hours of *Tom and Jerry*, 18 hours of *We Bare Bears*, 8 hours of *Mr. Bean*, and 46 hours of *Young Sheldon*. We standardize all videos by cropping out bottom text overlays (e.g., subtitles, credits) to prevent spurious language cues. Videos are segmented scene-by-scene into 5-second clips with the length of 81 frames, at 16 fps.

For each domain, we define the set of key characters: *Mr. Bean* (`Mr Bean`), *Tom and Jerry* (`Tom`, `Jerry`, `Spike`), *We Bare Bears* (`Ice Bear`, `Grizzly`, `Panda`), and *Young Sheldon* (`Sheldon`, `Missy`, `Mary Cooper`, `George Cooper`).

**Prompt:** *Panda is at a karaoke bar, singing loudly with his brothers, closing his eyes as if he were a superstar on TV.*

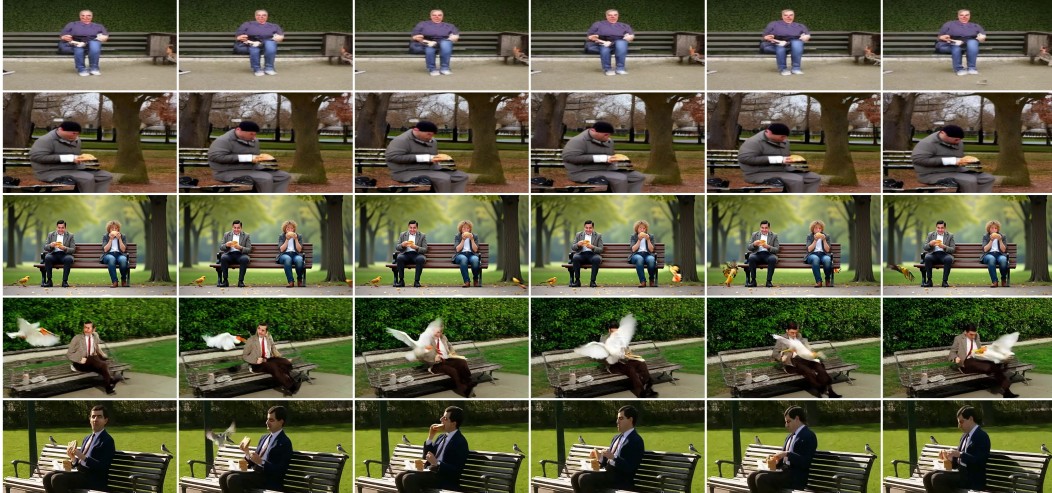

**Prompt:** *Mr. Bean is sitting alone on a park bench, trying to eat his sandwich while a bird keeps stealing the crumbs, making him look frustrated but also funny.*

Figure 5: **Comparison on single-subject generation.** From top to bottom: results from VideoBooth (Jiang et al., 2024), DreamVideo, Wan2.1-I2V (Team Wan, 2025), SkyReel-A2 (Fei et al., 2025) and ours.

## 4 EXPERIMENTS

### 4.1 BENCHMARKS

We evaluate our method using a comprehensive set of metrics along two dimensions. For overall video quality and temporal coherence, we adopt Consistency, Motion, Dynamic, Quality, and Aesthetic from VBench (Huang et al., 2024). To assess character-level consistency and interaction, we employ a vision–language model (VLM) and introduce four specialized metrics: Identity-P, Motion-P, Style-P, and Interaction-P. Specifically, we leverage Gemini-1.5-Flash (Team, 2024a) as the VLM backbone for these evaluations.

**Video Quality and Temporal Consistency.** (1) **Consistency** evaluates the overall video-text consistency across frames computed by ViCLIP (Wang et al., 2023). (2) **Motion** measures the level of smoothness of generated motions. (3) **Dynamic** quantifies the degree of motion dynamics using RAFT (Teed & Deng, 2020). (4) **Quality** measures the imaging quality, referring to the distortion (e.g., over-exposure, noise, blur) by the image quality predictor MUSIQ (Ke et al., 2021). (5) **Aes-**

**Prompt:** *Tom plays piano loudly. **Jerry** dances on the keys. **Mr. Bean**, wearing earmuffs with his suit, tries to conduct them like an orchestra. It turns into noisy chaos. The scene is cartoon style.*

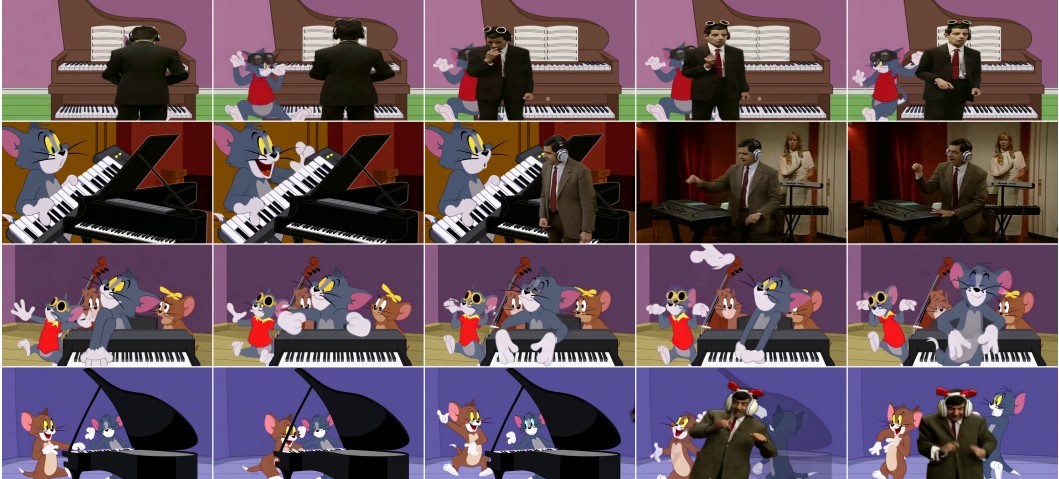

Figure 6: **Ablation on caption format.** From top to bottom: without tags, with `[character]` tag, with `[scene-style]` tag, and with both `character` and `scene-style` tags.

**thetic** evaluates the artistic and beauty value perceived by humans towards each video frame using the LAION aesthetic predictor ().

**Character Consistency and Interaction.**

**Identity-P** evaluates how well the generated video preserves each character's visual identity and distinctive features. The VLM assesses facial feature consistency, body proportions, characteristic attributes (e.g., Jerry's mouse ears, Tom's whiskers), and overall color scheme. A score of 10 indicates perfect identity preservation, where the character is immediately recognizable; a score of 1 indicates the character is completely unrecognizable.

**Motion-P** measures the authenticity of character-specific movements and behaviors relative to their canonical personality traits. The evaluation considers motion patterns (e.g., Jerry's quick scurrying, Tom's exaggerated sneaking), behavioral consistency, and expression of personality through movement. The VLM analyzes temporal sequences to assess alignment with the character's known behavior patterns.

**Style-P** assesses the consistency of each character's original artistic and visual style. This includes animation style (e.g., cartoon vs. realistic), aesthetic coherence with the source material, art direction fidelity, and rendering consistency. The VLM compares the generated video to its learned representation of the character's canonical appearance and stylistic conventions.

**Interaction-P** evaluates the naturalness and plausibility of multi-character interactions. The assessment considers spatial relationships, timing and coordination, believability of reactions and responses, and physical dynamics between characters. For single-character videos, this metric evaluates interactions with the environment and scene elements.

### 4.2 COMPARISON

We benchmark our method against state-of-the-art video generation baselines, including two single-subject customization approaches—VideoBooth (Jiang et al., 2024) and DreamVideo (Wei et al., 2024)—as well as foundation image-to-video models Wan2.1-I2V (Team Wan, 2025) and SkyReels-A2 (Fei et al., 2025), both of which support single- and multi-character generation. For multi-subject customization specifically, we compare directly against SkyReels-A2. Note that Wan2.1-I2V cannot directly generate videos from a character image. Thus, we first employ OmniGen (Xiao et al., 2025) to synthesize an image using the prompt and reference, which is then used as input to Wan2.1-T2V.

For single-subject evaluation, we generate 50 videos featuring 10 characters—five from cartoons (`Tom`, `Jerry`, `Grizzly`, `Ice Bear`, `Panda`) and five from live-action series (`Mr. Bean`,

**Prompt:** *Mr. Bean* rides a unicycle. *Ice Bear* juggles. *Panda* plays trumpet. The stage explodes fire.

Figure 7: **Ablation on augmentation data ratio.** From top to bottom: 0%, 5%, 10%, and 20% augmentation.

Table 1: **Comparison of recent video generation models across evaluation dimensions.** The first group of columns includes automatic evaluation metrics, while the last three report human evaluation scores. **Bold** indicates the best performance per column.

| Methods | Subject | VBench Metrics (Huang et al., 2024) | | | | | VLM Metrics | | | |
|---|---|---|---|---|---|---|---|---|---|---|
| | | Consistency | Motion | Dynamic | Quality | Aesthetic | Identity-P | Motion-P | Style-P | Interaction-P |
| VideoBooth (Jiang et al., 2024) | Single | 0.1287 | 0.9780 | 0.5094 | 0.6413 | 0.4896 | 4.45 | 3.72 | 5.43 | 4.44 |
| DreamVideo (ByteDance Team, 2025) | Single | 0.1851 | 0.9564 | | 0.6270 | 0.5002 | 4.51 | 4.16 | 6.82 | 5.37 |
| Wan2.1-I2V (Team Wan, 2025) | Single | 0.0682 | 0.9827 | 0.6530 | 0.7192 | 0.5857 | 5.27 | 5.10 | 7.94 | 6.41 |
| SkyReels-A2 (Chen et al., 2025a) | Single | 0.1469 | 0.9782 | 0.7843 | 0.7225 | 0.5850 | 6.17 | 4.55 | 7.82 | 6.78 |
| **Ours** | **Single** | **0.1893** | **0.9836** | **1.0000** | 0.5763 | **0.5967** | **6.12** | **5.41** | **8.06** | **7.24** |
| SkyReels-A2 (Chen et al., 2025a) | Multiple | 0.1314 | 0.9650 | 0.9787 | 0.7140 | 0.5371 | 6.17 | 4.55 | 6.28 | 4.94 |
| **Ours** | **Multiple** | **0.1833** | **0.9842** | **0.98555** | 0.6855 | **0.5813** | **6.48** | **5.50** | **7.26** | **5.22** |

`Sheldon`, `George`, `Mary`, `Penny`). For multi-subject evaluation, we generate 50 videos, each featuring 2–3 characters interacting within the same scene (noting that SkyReels-A2 supports fewer than three characters). These interactions span a wide range of scenarios, including inter-style (cartoon with real-life), intra-style (within cartoons or within real-life), inter-series (across different shows), and intra-series (within the same show). All reference images are included in the Appendix.

Figure 5 shows the qualitative comparison on single subject video generation. VideoBooth (Jiang et al., 2024) and DreamVideo (Wei et al., 2024) fail to preserve the visual identity of the reference character. SkyReel-A2 (Fei et al., 2025) and Wan2.1-I2T (Team Wan, 2025) retain identity to some extent—though they struggle with facial details—but fail to synthesize character-faithful motions. In contrast, our method consistently preserves visual fidelity and generates character-faithful motion.

Figure 4 presents qualitative comparisons on multi-character interaction, both within the same style and across different styles. While SkyReel-A2 (Fei et al., 2025) can synthesize acceptable single-subject videos, it struggles with complex interactions across multiple characters, especially in multi-style settings. Although it can place several characters into a shared scene, the results often exhibit visual inconsistencies and unnatural interactions. In contrast, our method enables contextually coherent interactions without compromising character identity or native style.

Quantitative results across nine metrics are reported in Table 1. Our method consistently outperforms prior approaches in both single- and multi-subject settings, demonstrating stronger identity preservation, faithful motion synthesis, and coherent style maintenance across diverse interaction scenarios. Additional comparison results are provided in the Appendix.

Table 2: **Effect of Caption Formatting.** We compare different caption formats, with and without structured tags, across multiple evaluation dimensions. Our full formatting with both `[character]` and `[scene-style]` tags achieves the best performance.

| Caption Format | VBench Metrics (Huang et al., 2024) | | | | | VLM Metrics | | | |
|---|---|---|---|---|---|---|---|---|---|
| | Subject-C | Background-C | Motion-S | Dynamic | Quality | Identity-P | Motion-P | Style-P | Interaction-P |
| No Tag (Baseline) | **0.8892** | **0.9136** | **0.9812** | **1.0000** | 0.6535 | 7.02 | 5.90 | 6.83 | 4.28 |
| w/o `[scene-style]` | 0.8668 | 0.8980 | 0.9754 | **1.0000** | 0.6353 | 7.31 | 5.80 | **6.95** | 4.47 |
| w/o `[character]` | 0.8758 | 0.9101 | 0.9759 | **1.0000** | **0.6938** | 7.33 | 5.42 | 6.80 | 4.47 |
| w/ Both (Ours) | 0.8530 | 0.8997 | 0.9747 | **1.0000** | 0.6588 | **7.35** | 5.80 | **6.95** | **5.30** |

Table 3: **Effect of Synthetic Data Augmentation.** We vary the proportion of synthetic videos relative to the original dataset and evaluate across the same dimensions used in Table 1.

| Ratio | VBench Metrics (Huang et al., 2024) | | | | | VLM Metrics | | | |
|---|---|---|---|---|---|---|---|---|---|
| | Subject-C | Background-C | Motion-S | Dynamic | Quality | Identity-P | Motion-P | Style-P | Interaction-P |
| 5% | 0.8739 | 0.9082 | 0.9826 | 0.9000 | 0.6836 | 7.67 | 6.03 | 7.15 | **4.83** |
| 10% | **0.8812** | **0.9151** | **0.9853** | 0.9500 | **0.6955** | **8.33** | **6.72** | **7.33** | 4.78 |
| 20% | 0.8442 | 0.8905 | 0.9779 | **1.0000** | 0.6728 | 8.30 | 7.10 | 7.08 | 3.90 |

## 4.3 ABLATION STUDY

We conduct ablation studies to investigate how different captioning strategies influence the model's ability to learn and ground each character as a distinct concept. Additionally, we analyze how varying the ratio of augmentation data affects the mitigation of style delusion.

**Captions Formats.** To evaluate the role of structured captions, we compare models trained with standard free-form captions against those trained with captions augmented by our proposed tags `[scene-style]` and `[character]`. The structured format provides explicit grounding of both scene attributes and character identities. Figure 6 illustrates a representative example of our video–caption pairs, showing how the tags enable more faithful alignment between visual entities and textual descriptions. The second row (without the `[scene-style]` tag) shows a shift from a cartoon scene to a realistic one. The third row (without the `[character]` tag) fails to include Mr. Bean in the generated video. In contrast, our method, using both tags, produces consistent scenes and correctly includes all characters.

**Augment Data Ratio.** We analyze the effect of incorporating our composited dataset (Section **??**) at different mixing ratios. Specifically, we vary the proportion of synthetic videos relative to the original curated dataset and evaluate the impact on performance. This experiment highlights how synthetic cross-character interactions contribute to improved generalization and robustness in multi-character video generation. Figure 7 illustrates a representative example. Both the baseline (0% augmentation) and the 5% augmentation setting generate a cartoon-style scene. While the 5% setting correctly produces a cartoon-style Mr. Bean, the 0% setting fails to preserve identity, instead generating a random cartoon character—possibly resembling one from the *We Bare Bears*,series. At 10% augmentation, the model successfully generates a realistic Mr. Bean with plausible interaction within the scene. However, at 20%, although Mr. Bean's appearance remains realistic, the interaction becomes less coherent, likely due to the overuse of synthetic data.

## 5 DISCUSSION

Despite the effectiveness of our framework in enabling controllable multi-character video generation, it comes with several limitations. Most notably, our approach relies on explicit identity annotations and LoRA fine-tuning. As a result, introducing a new character—whether from a different show or an unseen domain—requires retraining or fine-tuning the model. This limits scalability in open-world settings, where users may wish to generate videos with arbitrary or user-defined characters.

Furthermore, while our captioning and augmentation strategies mitigate style delusion and enable robust character disentanglement, the model still exhibits occasional failure cases in highly complex interaction scenes, especially when multiple characters with overlapping appearances or motion patterns are present.

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
