# OpenReview forum: "Character Mixing for Video Generation"
_ICLR.cc/2026/Conference — Submitted to ICLR 2026_

### Official Review · Reviewer_pamK · 2025-10-21

**Soundness:** 3
**Presentation:** 3
**Contribution:** 2
**Rating:** 6
**Confidence:** 3

**Summary:**

This paper introduces a new framework for enabling natural interactions between multiple characters from different visual domains, such as mixing Mr. Bean with Tom and Jerry using text-to-video (T2V) generation. The authors address two key challenges: non-coexistence, where characters from separate shows never appear together in training data, and style delusion, where mixed-style characters lose their original visual fidelity. To overcome these, they propose Cross-Character Embedding (CCE), which learns disentangled identity and behavioral representations from multimodal sources through structured character–action captions, and Cross-Character Augmentation (CCA), which generates synthetic co-existence data by compositing characters across domains while preserving native styles. As a result, the paper shows improvement compared to the baselines.

**Strengths:**

A key strength of this paper lies in its clear motivation, to bridge the gap between distinct fictional universes by enabling natural, style-consistent multi-character interactions. This being imaginative and well-grounded in practical generative modeling challenges. The authors present a thoughtfully designed data composition strategy, curating a dataset that balances diversity across cartoons and live-action domains while maintaining detailed character and style annotations.

This well-structured dataset, combined with the Cross-Character Embedding and Augmentation techniques, provides a solid foundation for learning both behavioral and stylistic nuances. Overall, the study is well-organized, and the presentation neatly ties together motivation, method, and results into a coherent and convincing.

**Weaknesses:**

[Major]
While the paper presents an impressive dataset and compelling demonstrations, its technical contribution beyond data construction and fine-tuning remains relatively incremental. The proposed Cross-Character Embedding (CCE) and Cross-Character Augmentation (CCA) modules, though effective, largely extend existing ideas of prompt-based disentanglement and synthetic compositing rather than introducing fundamentally new architectural or generative mechanisms.

The framework heavily relies on caption engineering, LoRA-based adaptation, and GPT-assisted annotation which approaches that are conceptually straightforward and depend more on large-scale data quality than algorithmic novelty. Consequently, while the work excels in implementation and application scope, its methodological innovation is modest compared to the scale of its dataset and the strength of its empirical results.

[Minor]
Some references related to video personalization are missing. ToonCrafter [1] proposed a fine-tuning technique for adapting video diffusion models to general cartoon domains, while AnyMoLe [2] introduced a video fine-tuning framework for generating motion-consistent videos of a single character.

There is typo in L464 (Section ??).

[1] ToonCrafter: Generative Cartoon Interpolation via Diffusion Models. Liu et al., SIGGRAPH ASIA, 2024.

[2] AnyMoLe: Any Character Motion In-betweening Leveraging Video Diffusion Models. Yun et al., CVPR, 2025.

**Questions:**

How many A100 GPUs were used and how long does it took for training?

Are you planning to open-source the code and dataset?

---

> ### Author Response · Authors · 2025-11-20
> **Response to Reviewer pamK**
>
> We thank Reviewer pamK for the questions and feedback. Please find our responses below.
>
> **Novelty**
>
> Please kindly refer to **Movitation, Goals and Novelty** in [**`Common questions`**](https://openreview.net/forum?id=SaBwaLcHNZ&noteId=nLyfOvaPjt).
>
> ------
>
> **Open-source code and dataset**
>
> We will release the code and the video captions used in this work. However, the original video dataset cannot be released due to copyright restrictions.
>
>
> ------
> **Training Time**
>
> The model is trained on 8 80G A100 GPUs for 5 days. More details are included in Section 3.3.
>
> ------
>
> **Missing Reference**
>
> We have included the mentioned references in the revised version.
>
> ------
> **Typos**
>
> We have corrected all typos in the revision.

---

### Official Review · Reviewer_gpfc · 2025-10-25

**Soundness:** 2
**Presentation:** 3
**Contribution:** 2
**Rating:** 4
**Confidence:** 3

**Summary:**

The authors identify two core challenges: non-coexistence, meaning some characters never appear together in training data, and style delusion, meaning cartoon and live action styles bleed into each other. To address this, they propose Cross Character Embedding, which uses structured captions of the form [character: name], action to disentangle identity and behavior, and Cross Character Augmentation, which pastes segmented characters into foreign style backgrounds to simulate cross-universe co-occurrence while preserving each character’s native look.

**Strengths:**

1. The paper tackles a concrete and highly visible capability gap in current video generation systems. Existing models can often render a single customized subject, but coherent multi-character interaction across different shows and even across cartoon versus live action domains remains very brittle.

2. The authors curate a reasonably large, annotated, behavior-aware dataset, and define evaluation metrics that target identity, motion, style, and interaction quality in multi-character settings.

3. Qualitative demos show convincing multi-character interactions that typical text-to-video systems struggle with.

**Weaknesses:**

1. Baseline fairness is underspecified. The proposed model is LoRA fine tuned on an 81 hours character specific dataset, whereas baselines may be evaluated mostly zero shot. This makes it hard to attribute the gains to CCE or CCA rather than just stronger task specific tuning.

2. Generalization is narrow. All results focus on about ten characters from four shows. The paper does not show that the approach scales to arbitrary new identities or unseen characters without new fine tuning.

**Questions:**

1. How were baselines adapted. Did you fine tune SkyReels A2, Wan2.1 I2V, or other baselines on the same eighty one hour dataset with LoRA style adapters, or are they evaluated zero shot. Please clarify to ensure a fair comparison.

2. Can you quantify failure modes. The discussion notes occasional breakdowns in highly complex multi character scenes with overlapping motion patterns.

---

> ### Author Response · Authors · 2025-11-20
> **Response to Reviewer gpfc**
>
> We thank Reviewer gpfc for the valuable questions and suggestions. Please find our responses below.
>
> ------
>
> **Baseline fairness comparison**
>
> We finetuned VideoBooth and DreamVideo, but did not fine-tune Wan-I2V and SkyReel-A2. We did not finetune SkyReel-A2 and Wan-I2V+OmniGen due to the extensive efforts required to process such training data. This clarification has been added to Section 4.2 in the revised version.
>
>
>
> ------
>
> **Failure cases visualization**
>
> We have included more discussion and visualization of failure cases in Section 5 and Figure .
>
> - Multi-Shot Inconsistency. When generating multi-shots videos, the characters' clothes may change across different shots. Please refer to this [video](https://mi-mi-x.github.io/src/assets/cross-shot-inconsistent.mp4) and [image](https://mi-mi-x.github.io/src/assets/cross-shot-inconsistent.png) for visual example, where both Tom's and Sheldon's outfits change between shots.
> - Missing Characters. When a text prompt includes too many characters, the model may fail to generate all of them.
>
> ------
>
> **Generalization to new characters**
>
> Our work specifically focuses on generalizing novel interaction between seen characters in new scenes and contexts. Generalizing to new identities without training is a misunderstanding of our core task (please also refer to **Personalization and Generalization Trade-off** in [**`Common questions`**](https://openreview.net/forum?id=SaBwaLcHNZ&noteId=nLyfOvaPjt)). The MIMIX problem setting is not about generating a new identity, rather, it is about building a rich, characteristically consistent, behaviorally grounded model of specific individuals, learned from extensive multimodal data. A single image or a short video is fundamentally insufficient to capture a person’s identity essence, behavioral style, or expressive dynamics. Therefore, generalization to unseen identities is outside the scope of our formulation and is conceptually distinct from the personalization problem we target.

---

### Official Review · Reviewer_UQK6 · 2025-10-31

**Soundness:** 2
**Presentation:** 2
**Contribution:** 2
**Rating:** 4
**Confidence:** 4

**Summary:**

This paper introduces a new task termed Character Mixing, aiming to generate multi-character videos where characters from different IPs or styles appear and interact in the same scene. The authors construct a dataset by mixing existing videos and captions with the help of large language and vision models, and design two modules (CCE and CCA) to align cross-character and cross-style information. Experiments on several cartoon characters are reported.

**Strengths:**

1. The idea of character mixing is creative and intuitively interesting, which could inspire further exploration in cross-style or cross-domain video generation.

2. The authors contribute a dataset with structured captions for multi-character interaction videos, which may be useful for future research.

**Weaknesses:**

1.Low problem significance.
While entertaining, the problem does not address a clear or impactful research challenge. It is more of a creative application than a fundamental scientific question. The motivation for why character mixing matters for the video generation community is weak.

2.Limited methodological novelty.
The proposed CCE and CCA modules mainly rely on prompting and data augmentation rather than introducing new modeling or learning principles. The improvements largely stem from the underlying base model's(Wan2.1 14B used, smaller size or other open-source model should be presented to support model-independent claim)  capability rather than the proposed method itself.

3.No formalization or mathematical clarity and training details.
The paper lacks any formal task definition, notation, or training objective. Without clear training details, it is difficult to reproduce and evaluate the soundness of the approach.

4.Limited scalability and generalization to new characters.
The reliance on explicit character-level annotation and LoRA-based fine-tuning means adding a new character (especially from unseen domains) requires re-training or substantial data preparation (highlighted in Discussion, Page 9). This approach does not scale gracefully for open-world or user-specified characters.

5.Missing analysis and discussion.
There is little exploration of failure cases, interaction quality (e.g., temporal consistency, occlusion handling), or computational cost. The paper reads more like a demo than a scientific study.

**Questions:**

1.Can the method handle unseen characters or styles at inference time?

2.What would happen if GPT-4o/Gemini were replaced with open-source models (e.g., LLaVA, Qwen2-VL)?

3.How is the proposed CCE/CCA architecture implemented and trained?

---

> ### Author Response · Authors · 2025-11-20
> **Response to Reviewer UQK6**
>
> **Generalization and scalability**
>
> Our generalization ability does not lie in creating new characters, but in generating new cross-character interactions in new scenes. Please refer to  [**`Common questions`**](https://openreview.net/forum?id=SaBwaLcHNZ&noteId=nLyfOvaPjt) for more details.
>
> ------
>
> **CCE/CCA implemented and training details?**
>
> The CCE is a prompting mechanism, and CCA is an data augmentation mechanism, as clearly described in Section 3.1 and 3.2. We respectfully encourage the reviewer to refer to these sections for more details.
>
> ------
>
> **Problem Significance and Novelty**
>
> Please refer to **Movitation, Goals and Novelty** in [**`Common questions`**](https://openreview.net/forum?id=SaBwaLcHNZ&noteId=nLyfOvaPjt) for more discussion.
>
> We emphasize that the `style-delusion` problem is both common and critical. Even the state-of-the-art method Nano-Banana fails when mixing characters with different visual styles, as illustrated in [this example](https://raw.githubusercontent.com/mi-mi-x/mi-mi-x.github.io/refs/heads/main/src/assets/nano-banana-example.jpg).
>
> ------
>
> **Misunderstanding: improvements largely stem from the base model rather than the proposed method**
>
> - **Baed model without character concept.** Without any training, the base Wan-T2V model cannot generate any of the 10 characters mentioned in the paper.
> - **Based model with finetuning still suffers from style-delusion.** Withou our CCA, the model still exhibits style-delusion problem, producing cartoonish Mr. Bean or realistic Ice Bear, as shown in Figure 2.
> - **Model scale ablation.** We trained our method on smaller Wan2.1-T2V-1.3B and Wan2.2-T2V-5B model, and all models can effectively mix different style characters. It shows that the mixing ablilaty originates from our approach, while the video quality of different models varies depending on the foundation model. Please refer to **Model-Agnostic Validation** in `Common question` for more details and visual comparison.
>
> In summary, these indicate that the improvements do not arise from the base model alone but from our approach. We respectfully encourage the reviewer to revisit the relevant sections and figures for a more detailed understanding.
>
> ------
>
> **Formalization or mathematical clarity**
>
> As stated in the paper, our model is trained as a LoRA on Wan2.1-T2V (Section 3.3 and Figure 3). The training objective follows the Flow-Matching Loss, consistent with Wan2.1’s formulation. We assume the reviewer is familiarity with LoRA training and kindly recommend revisiting the corresponding sections for further clarity.
>
> ------
>
> **Failure Cases**
>
> We have added further discussion and visualizations of failure cases in Section 5.
>
> - **Multi-Shot Inconsistency**. When generating multi-shots videos, the characters' clothes may change across different shots. Please refer to this [video](https://mi-mi-x.github.io/src/assets/cross-shot-inconsistent.mp4) and [image](https://mi-mi-x.github.io/src/assets/cross-shot-inconsistent.png) for visual example, where both Tom's and Sheldon's outfits change between shots.
> - **Missing Characters**. When a text prompt contains too many characters, the model may omit one or more, instead of generating all of them.

---

> ### Author Response · Authors · 2025-11-20
> **GPT-4o/Gemini were replaced with open-source models**
>
> We tested two open-source models Qwen3-VL-235B-A22B and Intern-VL3.5-241B-A28B on our dataset for video captioning. **Both models successfully recognize the main characters** and generate plausible captions as shown below for the [input video](https://mi-mi-x.github.io/src/assets/516444653-f99fd0e9-1ee0-40ba-9569-c02af7a4c2cd.mp4).
>   - **GPT-4o**: In this clip from We Bare Bears, the three brothers — Grizzly, Panda, and Ice Bear — are seen navigating a comical situation. Panda appears anxious and expressive, reacting dramatically as something unexpected happens. Grizzly, the upbeat and energetic oldest brother, tries to handle the chaos with enthusiasm, while Ice Bear, ever calm and stoic, moves with silent efficiency. The brothers seem to be engaged in an amusing adventure or mishap, showcasing their distinct personalities: Panda’s sensitivity, Grizzly’s boldness, and Ice Bear’s quiet competence. The animation reflects the show’s signature mix of humor, sibling dynamics, and heartwarming moments.
>   - **Qwen3-VL**: In a cozy diner booth, the We Bare Bears—Panda, Ice Bear, and Grizzly—are seated together. Panda, on the left, looks downcast while holding a drawing. Ice Bear, in the middle, sips his lemonade with a calm expression. Grizzly, on the right, appears frustrated, reading a newspaper titled "PARK VACATION" and gesturing angrily. A waitress stands by their table, looking annoyed. The scene captures a moment of tension among the trio, with Grizzly’s irritation contrasting with Ice Bear’s serenity and Panda’s sadness, all under the watchful eye of the exasperated server.
> - **Intern-VL**: In this scene from "We Bare Bears," three anthropomorphic bears—Panda, Ice Bear, and Grizzly—are seated in a diner. Grizzly, on the right, looks concerned as he holds a pamphlet. Panda, on the left, appears to be showing Ice Bear, who is in the middle, a drawing on a napkin. They each have drinks with lemon slices. In the background, other diners are visible, including a woman with red hair. The setting is casual, with various signs and license plates decorating the walls. Grizzly seems to be explaining something important, as indicated by his expression and the pamphlet marked with a red "B." The atmosphere is light-hearted yet slightly tense due to Grizzly's concern.

---

### Official Review · Reviewer_Fsis · 2025-11-01

**Soundness:** 3
**Presentation:** 3
**Contribution:** 4
**Rating:** 6
**Confidence:** 4

**Summary:**

This paper introduces a novel framework for generating videos that feature multiple characters from different fictional universes (e.g., cartoons and live-action shows) interacting with each other. The authors identify two primary challenges: 1) the "non-coexistence" of characters in training data, and 2) "style delusion," where characters' visual styles blend undesirably. To address these, they propose a two-part solution. First, **Cross-Character Embedding (CCE)**, a prompt engineering strategy that uses structured captions of the form `` `[Character: <name>], <action>` `` to disentangle character identity and behavior. This allows the model to learn character-specific traits from their respective source videos and compose them at inference time. Second, **Cross-Character Augmentation (CCA)**, a data augmentation technique that synthetically creates training examples of cross-style interaction by segmenting characters from one domain and pasting them into scenes from another. These augmented clips are captioned with an additional `` `[scene-style: <style>]` `` tag to help the model preserve stylistic integrity. The authors fine-tune a large-scale text-to-video model (Wan2.1) on a curated 81-hour dataset and demonstrate through extensive experiments that their method significantly outperforms existing baselines in identity preservation, interaction quality, and style consistency. They also introduce a new benchmark and a set of VLM-based evaluation metrics for this specific task.

**Strengths:**

*   **Problem Formulation and Significance**: The paper formulates a compelling and significant research problem: enabling characters from different "universes" to interact naturally in generated videos. This is a natural evolution of personalized generation and has high potential for creative applications.
*   **Novel Methodology**: The proposed framework is original in its combination of two distinct ideas to solve two well-defined problems. CCE (via prompt structure) tackles the non-coexistence of characters, and CCA (via synthetic compositing) addresses the style delusion problem. This two-pronged approach is elegant and shown to be effective.
*   **Strong Empirical Results**: The qualitative results are visually impressive and clearly demonstrate the superiority of the proposed method over existing approaches, which either fail to maintain identity or cannot produce coherent interactions. The quantitative results, despite the aforementioned issues with the ablation tables, generally show a strong performance lead, especially on the task-specific VLM metrics.
*   **Benchmark Contribution**: The introduction of a dedicated benchmark for multi-character interaction, including a new suite of VLM-based metrics tailored to character identity, motion, style, and interaction, is a substantial contribution in its own right. It provides a more meaningful way to evaluate models on this task than standard metrics alone.

**Weaknesses:**

*   **Scalability and Data Dependency**: The method's primary weakness is its reliance on fine-tuning using a large corpus of video data for a pre-defined set of characters. As implied in Section 3.3, each new character universe (e.g., a TV show) requires collecting hours of video footage and undergoing an expensive fine-tuning process. This makes it difficult to scale to new, arbitrary characters in an open-world setting. As acknowledged by the authors, this is a significant limitation.
*   **Lack of Human Evaluation and Unjustified Metric Choices**: For a task where success is highly subjective (e.g., "authenticity", "plausibility"), the absence of a human study is a major weakness. The paper instead introduces VLM-based metrics but fails to justify why established identity preservation metrics (e.g., face recognition similarity) were not used or compared against, especially for the human-like characters. While VLM evaluation is innovative, its reliability is questionable without proper protocol.
*   **Questionable Reproducibility of VLM-based Evaluation**: The use of a VLM as a core evaluation tool introduces significant reproducibility concerns. The output of VLMs can be stochastic due to parameters like `temperature`. The paper fails to describe the protocol used to ensure deterministic and reproducible scores. Key details are missing: Was `temperature` set to 0? Were scores averaged over multiple runs? What were the exact prompts used? Without this information, the quantitative results in Table 1, 2, and 3 are not fully credible.
*   **Unverified Claim of Model-Agnosticism**: The paper claims its framework is "model-agnostic" (Section 3.1) but provides no empirical evidence. All experiments are conducted by fine-tuning a single base model (Wan2.1-T2V-14B). Without applying the CCE and CCA framework to at least one other distinct T2V backbone, this claim of generalizability remains unsubstantiated.
*   **Clarity and Consistency of Ablation Studies**: The quantitative results in the ablation section (Tables 2 and 3) are poorly presented. There are numerical inconsistencies for what should be identical experimental conditions across different tables. For example, the results for the full model ("Ours" in Table 1, "w/ Both" in Table 2, and "10%" in Table 3) all report different scores. This makes it impossible to confidently assess the individual contributions of the proposed components. This must be fixed.

**Questions:**

1.  **On Data Requirements**: Could you please clarify the data requirements more explicitly? The paper implies a single fine-tuning on a mixed dataset. Does this mean to add a new character from a new TV show, one must re-run the entire fine-tuning process on the combined old and new data? What is the approximate training time (e.g., in GPU-hours) for the reported 5-epoch fine-tuning on the 81-hour dataset?
2.  **On Evaluation Metrics**:
    *   (a) Could you justify the decision to exclusively use VLM-based metrics for identity preservation over established methods like face recognition ID similarity, at least for the human characters (Mr. Bean, Young Sheldon's cast)? A comparison showing the VLM's superiority would strengthen your metric choice.
    *   (b) Crucially, what protocol was used to ensure the VLM evaluation is deterministic and reproducible? Please provide the exact prompts, model version, and parameter settings (especially `temperature`) used to query Gemini-1.5-Flash for the Identity-P, Motion-P, Style-P, and Interaction-P scores in the appendix.
3.  **On Generalizability**: To substantiate the "model-agnostic" claim, could you provide any results or insights, even preliminary, from applying your CCE and CCA framework to a different base T2V model?
4.  **On Inconsistent Ablation Results**: Could you please clarify and unify the results presented in Tables 1, 2, and 3? Specifically, please ensure that the results for the same experimental setup are consistent across all tables and explain the counter-intuitive findings in Table 2 (e.g., the drop in `` `Subject-C` ``).
5.  **On CCE's Mechanism**: The term "Cross-Character Embedding" suggests a specific learned representation. Can you clarify whether your method learns an explicit, separable embedding for each character, or if CCE is more accurately described as a structured prompting technique that influences the text-conditioning of the frozen T2V model?

---

> ### Author Response · Authors · 2025-11-20
> **Response to Reviewer Fsis**
>
> We thank Reviewer Fsis for the valuable questions and suggestions regarding the evaluation. Please find our responses below.
>
> ------
>
> **Data Requirements and training for new characters**
>
> Yes, it requres continued training for new characters using combined old and new data. The model can be initialized from the pre-trained model weights rather than trained from scratch. To accelerate the process, a biased sampling strategy can be employed, where new data are sampled more frequently than old data.
>
> ------
>
> **Why use VLM metrics instead of ID Similarity**
>
> We did not adopt this approach for two main reasons:
>
> - **Limited applicability of face recognition algorithms.** Such algorithms are primarily designed for real human faces and do not perform reliably on cartoon characters.
> - **Conceptual limitation of image-based similarity**: We purposely choose world-famous characters like *Mr. Bean* as our experiment sugjects so that humans can easily judge whether a generated video resembles Mr. Bean without any reference videos. So dose a vision-language model that already knows the person. Reference-based similarity, however, is akin to identifying a stranger from a single photo—without any prior knowledge of their diverse appearances, expressions, or characteristic motions—and thus often depends on superficial cues such as clothing color. In contrast, a Vision-Language Model (VLM) possesses prior knowledge about such well-known characters, enabling evaluation that extends beyond pixel-level similarity to capture facial identity and behavioral consistency.
>
> From our perspective, both VLM and image-based ID similarity metrics have their respective advantage. Image-based similarity mainly focus on measuring static appearance (mainly face), whereas VLM-based evaluation provides a more comprehensive assessment of dynamic appearance, characteristic motion, full-body consistency, and overall semantic alignment.
>
> ------
>
> **VLM Metrics Implement Details**
>
> We employ Gemini-1.5-Flash with a temperature of 0.3, top-p of 0.95, and top-k of 40. The code for computing the metrics can be downloaded [here](https://mi-mi-x.github.io/src/assets/gemini_video_scorer.py). We have included details including the VLM version, configuration, and guidance prompts in the supplementaty Section C.
>
> ------
>
> **Generalizability and Model-Agnosticism**
>
> Please refer to [**`Common questions`**](https://openreview.net/forum?id=SaBwaLcHNZ&noteId=nLyfOvaPjt) for more discussion.
>
> ------
> **Inconsistent Ablation Results**
>
> Tables 1, 2, and 3 are not directly comparable because Table 1 reports results trained on the full dataset with four casts, whereas the ablation studies use only two casts: Table 2 uses Mr. Bean and Tom & Jerry, and Table 3 uses Mr. Bean and We Bare Bears.
>
> The subject-C score is computed based on the DINO feature similarity across frames for individual subjects (e.g., a person, a cat). The drop in subject-C primarily because **DINO struggles to extract reliable features from these generated mixed-style images**.
>
> We will unify table 1,2 and 3 under the same setting using all data, and the revised version will be updated once the experiments are completed.
>
> ------
> **CCE's Mechanism**
>
> We do not learn a separable embedding for each character. CCE functions more as a structured prompting mechanism rather than an explicitly learned embedding space.
>
> ------
> **Human-Evaluation Metrics**
>
> We are conducting the user study and will update the results once it is completed.

---

### Author Response · Authors · 2025-11-20
**Common questions**

We thank the reviewers for the valuable feedback. Please find the answers of the common questions below.

---------------
**1. Movitation, Goals and Novelty**

Our **motivation** stems from a simple yet profound question: *Can we faithfully recreate a real person—or even a stylized cartoon character—by extensively learning from their videos and scripts?* Existing generative models can produce videos that look like a person, but what we seek is to create videos that feel like them: capturing their essence, expressions, motion patterns, personality cues, and how they would react when placed in new environments or interacting with characters they have never seen. We refer to this capability as `personalization`.

Inspired by this, we started our research journey across four stages:

| **Stage**  | **Train Data** | **Challenge and Focus**  |
|------------|---------------------|-------------------|
| **single-person** | *Mr. Bean* | **`Personalization`**: preserving the identity, motion and behavior of an individual character |
| **cross-person** | *Three Bears* | **`Multi-characters grounding`**: enable accurate multi-character grounding and caption, allowing the video model to asccociate each character with its visual appearance. |
| **cross-cast, single-style** | *Three Bears* *Tom & Jerry* | **`Non-coexistence`**: enable coherent interactions across characters from different cast. |
| **cross-cast, cross-style** | *Mr. Bean*  *Tom & Jerry*| **`Style-Delusion`**: enable coherent interactions among characters of different styles while preserving their identity and visual appearance.|

In summary, our **goal** is to **learn personalized characters and enable cross-cast and cross-style interactions**.

Our **novelty** lies in addressing this goal and overcoming the associated challenges, as none of the existing works are capable of achieving this. Most related studies mainly focus on a single subject, a single style, or a single motion personalization only.

---------
**2. Generalization and Personalization Trade-off**

Our approach requires training on unseen characters. But **generalization and personalization are inherently a trade-off**. General video models exhibit strong generalization ability but limited personalization, whereas personalized models produce videos with highly preserved identity but require additional training. Its difficult to achieve both simultaneously. **Our approach, primarily emphasizing personalization, therefore involves training on unseen characters** to acquire the necessary visual and motion knowledge.

We have included this discussion in Section 5 in the revised version.

---------

**3. Our Generalization: New Interactions in New Scenarios**

Our work specifically focuses on **generalizing a single person’s reactions to novel scenes and novel contexts**. Some reviewers asked about generalizing to new identities without training, but this reflects a misunderstanding of our core task. The MIMIX problem setting is not about generating a generic identity from a few images; rather, it is about building a rich, characteristically consistent, behaviorally grounded model of specific individuals, learned from extensive multimodal data. A single image or a short video is fundamentally insufficient to capture a person’s identity essence, behavioral style, or expressive dynamics. Therefore, generalization to unseen identities is outside the scope of our formulation and is conceptually distinct from the personalization problem we target.

We have included this discussion in Section 5.

------
**4. Model-Agnostic Validation**

We trained our approach on **Wan2.2-TI2V-5B**. Here is a comparison of [Wan2.2-5B](https://mi-mi-x.github.io/src/assets/wan2.2-5B-Mr._Bean_tries_to_eat_a_spaghetti_sandwich_on_a_be.mp4) and [Wan2.1-14B](https://mi-mi-x.github.io/src/assets/wan2.1-14B-Mr._Bean_tries_to_eat_a_spaghetti_sandwich_on_a_be.mp4). We observe that Wan2.1-14B produces higher visual quality and notably more stable and consistent motion than Wan2.2-5B. This performance gap may stem from Wan2.1-14B being nearly three times larger in size, and Wan2.2-5B employs higher compressed video VAE. **Both models can effectively mix cartoon and realistic character,** demonstrating that our approach is model-agnostic, where the video quality varies depending on the foundation model.

We will include this in the revised version.

------
**5. Training Time**

The model is trained on 8 80G A100 GPUs for 5 days. More details are included in Section 3.3.

---

### Meta-Review · Area_Chair_GnuX · 2026-01-06

**Summary:**

This paper presents a novel framework for multi-character text-to-video generation, addressing the challenges of non-coexistence and style delusion by introducing Cross-Character Embedding and Cross-Character Augmentation. It demonstrates the ability to generate natural interactions between characters from different visual styles while preserving their identities and motion patterns. Reviewers generally found the problem engaging and the results visually compelling, but raised concerns regarding scalability to new characters, reliance on fine-tuning and curated data, lack of human evaluation, reproducibility of VLM-based metrics, insufficient ablation study clarity, and limited methodological novelty beyond prompting and data augmentation.

**Reviewer Concerns:**

In the rebuttal, the authors addressed several reviewer concerns by clarifying that their method emphasizes personalization over generalization to unseen identities, explaining that VLM-based metrics were chosen over face recognition due to their broader applicability across cartoons and humans, and providing implementation details such as Gemini configuration and training specifics. They also committed to adding human evaluation and correcting inconsistencies in ablation tables. However, key issues remain unresolved, including the lack of evidence for model-agnostic claims (no experiments with other base models), limited scalability (requiring retraining for new characters), and unaddressed reproducibility concerns regarding VLM evaluation protocols. The methodological novelty is still viewed by some reviewers as incremental, relying heavily on prompting and data augmentation rather than fundamental algorithmic innovation.

**Reviewer Scores:**

Reviewers UQK6 and gpfc would likely maintain their scores around 4 (marginally below acceptance), as the authors' responses did not resolve their core concerns about limited scalability, the lack of experiments proving model-agnosticism, and the perceived incremental nature of the methodology. Reviewer Fsis, who initially gave a 6 (marginally above acceptance) but highlighted significant weaknesses in evaluation and reproducibility, might lower the score to a 4 or maintain a weak 6, given that the authors provided implementation details for VLM metrics and promised to fix table inconsistencies, but failed to substantiate the model-agnostic claim or fully address the reproducibility protocol. Reviewer pamK, who initially gave a 6, might raise the score to a 8, as they appeared more receptive to the problem's significance and were satisfied with the authors' clarifications on training details, references, and plans to open-source code. Thus, the post-rebuttal scores would reflect a split, with two reviewers likely below the acceptance threshold and two at or above it, leading to a borderline overall rating.

---

### Decision · Program_Chairs · 2026-01-26

Reject